# Modulation of Guanylate Cyclase Activating Protein 1 (GCAP1) Dimeric Assembly by Ca^2+^ or Mg^2+^: Hints to Understand Protein Activity

**DOI:** 10.3390/biom10101408

**Published:** 2020-10-05

**Authors:** Francesco Bonì, Valerio Marino, Carlo Bidoia, Eloise Mastrangelo, Alberto Barbiroli, Daniele Dell’Orco, Mario Milani

**Affiliations:** 1CNR-IBF, Istituto di Biofisica, Via Celoria 26, I-20133 Milan, Italy; francesco.boni@unimi.it (F.B.); carlo.bidoia@studenti.unimi.it (C.B.); eloise.mastrangelo@unimi.it (E.M.); 2Dipartimento di Bioscienze, Università di Milano, Via Celoria 26, I-20133 Milan, Italy; 3Dipartimento di Neuroscienze, Biomedicina e Movimento, Sezione di Chimica Biologica, Università di Verona, I-37134 Verona, Italy; valerio.marino@univr.it; 4Dipartimento di Scienze per gli Alimenti, la Nutrizione e l’Ambiente, Università degli Studi di Milano, Via Celoria 2, I-20133 Milan, Italy; alberto.barbiroli@unimi.it

**Keywords:** quaternary assembly, calcium-binding proteins, EF-hand, protein-protein interaction, small-angle X-ray scattering, molecular dynamics simulations, protein modeling, protein dynamics, size exclusion chromatography, multi-angle light scattering

## Abstract

The guanylyl cyclase-activating protein 1, GCAP1, activates or inhibits retinal guanylyl cyclase (retGC) depending on cellular Ca^2+^ concentrations. Several point mutations of GCAP1 have been associated with impaired calcium sensitivity that eventually triggers progressive retinal degeneration. In this work, we demonstrate that the recombinant human protein presents a highly dynamic monomer-dimer equilibrium, whose dissociation constant is influenced by salt concentration and, more importantly, by protein binding to Ca^2+^ or Mg^2+^. Based on small-angle X-ray scattering data, protein-protein docking, and molecular dynamics simulations we propose two novel three-dimensional models of Ca^2+^-bound GCAP1 dimer. The different propensity of human GCAP1 to dimerize suggests structural differences induced by cation binding potentially involved in the regulation of retGC activity.

## 1. Introduction

In photoreceptor cells, light photons are transformed into electric signals by the coordinate interplay of a complex network of proteins, regulated by different chemical messengers [1].

Guanylyl cyclase-activating protein 1 (GCAP1) belongs to the calmodulin superfamily and hosts 4 EF-hand motifs (EF1–EF4). At low Ca^2+^ concentration (<100 nM) GCAP1 EF2 can bind at least one Mg^2+^ ion [2] activating retinal guanylyl cyclase 1 (retGC1) [3,4]. Active retGC1, in turn, replenishes the second messenger cGMP, switching off the light-activated cascade and restoring the photoreceptors‘ dark state. High cGMP concentration increases the influx of Ca^2+^ through the cyclic nucleotide-gated (CNG) channels determining GCAP1 Ca^2+^-bound state (EF2-4 bind 3 Ca^2+^ ions) that inhibits retGC1 activity. During evolution, GCAP1 N-terminal EF-hand (EF1) lost the capability to bind calcium and became involved in the binding of retGC1 and modulation of its activity [5,6,7].

So far, 22 pathological point mutations of GCAP1 have been identified, which are associated with an autosomal dominant cone or cone-rod dystrophy [8]. Such mutations usually induce a lower affinity for Ca^2+^ determining the deregulation of retGC1 at physiological calcium levels, with progressive degeneration of the cone photoreceptors [9,10]. Nevertheless, exceptions exist for substitutions at the N-terminal domain (L84F and L176) that do not perturb the affinity for Ca^2+^ in the respective GCAP1 variants [11,12]. Recent work suggests also the involvement of GCAP1 in retinitis pigmentosa, a severe form of progressive rod-cone dystrophy [13].

It is known that purified GCAP1 displays both monomeric and dimeric species [14]. Recently the dimeric assembly raised particular interest for its possible involvement in retGC1 regulation [15,16]. The dimeric assembly proposed by Lim et al. [15] involves contacts between residues comprised in EF1 and EF2 motifs, i.e., in the portion of the protein that is thought to interact with retGC1, specifically His19, Tyr22, Phe73, and Val77. Mutations of these residues abolished the dimeric assembly as proven by analytical gel filtration [15].

In this work, we performed several experiments on recombinant human (h) GCAP1, to thoroughly analyze the dimeric assembly. Microscale thermophoresis, size exclusion chromatography (SEC) and small-angle X-ray scattering (SAXS) data confirmed the presence of a monomer-dimer equilibrium in solution influenced by Ca^2+^ or Mg^2+^ binding and by ionic strength.

Using the ZDOCK program [17] for rigid-body protein–protein docking, we produced a large number of possible dimeric assemblies, ranked using the empirical docking score, and filtered according to the fulfillment of distance constraints imposed by previous EPR double electron-electron resonance (DEER) experiments [15]. The best 3 dimers were selected for further characterization and optimization by molecular dynamics (MD) simulations and analyzed with SEC-SAXS data collected at the highest protein concentration. The optimized dimeric structures were used to predict the change in the free energy of binding (ΔΔG°) upon mutations, which are known to induce GCAP1 monomerization [15]. The two resulting three-dimensional models of dimeric GCAP1 in its Ca^2+^-bound form presented in this work, recapitulate all current experimental lines of evidence from us and other authors and might have implications on the regulatory activity of GCAP1.

## 2. Materials and Methods

### 2.1. Human GCAP1 Expression and Purification

To produce the N-terminal myristoylated hGCAP1, the recombinant plasmid pET11a carrying the cDNA sequence was co-transformed in the *E. coli* strain BL21-CodonPlus RP (Agilent Technologies, Santa Clara, CA, USA) together with a pBB131 plasmid carrying the coding sequence for yeast *N*-myristoyltransferase (NMT). The hGCAP1 construct has the E6S mutation needed for the post-translational myristoylation.

Cell cultures were grown at 37 °C in 2 L of LB medium containing 100 µg/mL ampicillin (Amp), 30 µg/mL kanamycin (Kan), and 34 µg/mL chloramphenicol (Cm). The myristic acid (220 µM) was added when the culture reached the OD600 = 0.4. Fifteen min later, the expression of hGCAP1 and NMT was induced by the addition of 1 mM IPTG. Cells were harvested after 3 h at 37 °C and frozen at –20 °C.

Typically, 3 g of *E. coli* cells were resuspended in lysis buffer (50 mM Tris-HCl pH 8.0, 50 mM NaCl, 10 µg/mL Deoxyribonuclease I, 20 mM MgSO_4_, 5 mM DTT, and protease inhibitors (cOmplete, Roche, Basel, Switzerland) and lysed by high-pressure cell disruptor (Basic Z Bench Top; Constant Systems Limited, Daventry, Northants, UK) at 25 KPSI. The crude extract was centrifuged at 38,700× RCF for 30 min and filtered through a 0.45 µM filter. The clarified soluble extract was loaded in a HiPrepTMQ XL 16/10 (GE Healthcare Life Sciences, Boston, MA, USA) in loading buffer (50 mM Tris-HCl pH 8.0, 50 mM NaCl, 2 mM EDTA, 5 mM DTT) and eluted with a linear gradient of NaCl (from 0.2 to 1 M).

Fractions corresponding to hGCAP1 were pooled, concentrated, and passed through a HiLoadTM 16/600 SuperdexTM75 pg (GE Healthcare Life Sciences) equilibrated with 20 mM Tris/HCl pH 8.0, 100 mM NaCl, 5 mM DTT, 1 mM CaCl_2_, and one cOmplete protease inhibitor tablet. All the purification steps were performed using an ÄKTA pure 25L (GE Healthcare) at 4 °C and analyzed with SDS-PAGE.

To obtain the apo form of hGCAP1, the protein was further purified using a hydrophobic interaction chromatography, decreasing the NaCl concentration from 300 mM to 0 mM. EGTA was added at 4 mM concentration both in the loading and the elution buffer.

### 2.2. Circular Dichroism

Far-UV Circular Dichroism (CD) spectra of hGCAP1 with and without Ca^2+^ were recorded in a Jasco J-810 spectropolarimeter (JASCO Europe, Cremella, Italy) equipped with a Jasco temperature controller module PFD-425S. All measurements were performed in a 0.1 cm path length quartz cuvette at 0.2 mg/mL protein concentration in 20 mM Tris-HCl, pH 7.5, 100 mM NaCl, 2 mM DTT, added with 2 mM Ca^2+^ or 2 mM EGTA. Far-UV spectra were recorded from 190 to 260 nm, at 20 and 95 °C (temperature was increased in ramp mode at 1 °C/min).

### 2.3. Analytical Size Exclusion Chromatography

To study the monomer-dimer equilibrium of the Ca^2+^-bound hGCAP1 the protein buffer was exchanged into 20 mM Tris-HCl pH 8.0, 100 mM NaCl, 1 mM DTT, 5 mM CaCl_2_ using a PD-10 Desalting Column (GE Healthcare). Calcium was substituted with 5 mM MgCl_2_ and 2.5 mM EGTA to study the Mg^2+^-bound protein.

Different protein dilutions were prepared (87 µM, 43 µM, 21 µM, 10 µM, 4 µM, 2 µM) and injected (200 μL) in a SuperdexTM75 Increase 10/300 GL column (GE Healthcare Life Science) with 0.8 mL/min flow rate.

Assuming a linear dependency of Ve from the logarithm of the Mw (Ve = A × log(Mw) + B) and assuming that in the presence of a monomer-dimer equilibrium the apparent Mw is related to the percentage of each species through the equation Mw = (%dim × 22.9 + 22.9) we get the relation Ve = A × log((%dim × 22.9 + 22.9)) + B. Describing the monomer-dimer equilibrium through the dissociation constant Kd we arrive at the 3 parameters (A, B, and Kd) equation to fit the Ve vs. concentration (c) curve:Ve = A·log (22.9·(1 − Kd/4c·( − 1 + (1 + 8c/Kd)^(0.5)^)) + 22.9) + B

### 2.4. Microscale Thermophoresis (MST)

The human GCAP1 was labeled with the red fluorescent dye NT-647-NHS with the Monolith NTTM Protein Labeling kit according to the manufacturer protocol (Nanotemper Technologies, München, Germany). The NT-647-NHS carries a reactive NHS-ester group that reacts with primary amines (lysine residues) to form a covalent bond, achieving about 1:1 ratio of labeled protein to dye. The dye solution was prepared at 30 µM in the supplied labeling buffer. GCAP1 (10 µM) was diluted in a 1:2 ratio with the labeling solution and incubated at room temperature for 30 min. Sixteen serial dilutions of unlabeled protein were prepared from 46.6 µM to 11.4 nM in 20 mM Tris-HCl, 100 mM NaCl, pH 7.5, 1 mM DTT, 5 mM CaCl_2_ supplied with 0.05% Tween20. Labeled hGCAP1 (10 nM) was added in each of the sixteen dilutions and the reactions were loaded in sixteen capillaries. After confirming that the fluorescence of each capillary was constant, the MST traces were recorded at 24 °C with 40% laser excitation power and 40% MST power. 

The same experiments were repeated using a 20 mM Tris-HCl, 150 mM KCl, 5 mM MgCl_2_, 2.5 mM EGTA, pH 7.5, 1 mM DTT buffer supplied with 0.05% Tween20 to study the magnesium-bound hGCAP1.

All the MST experiments were carried out with a Monolith NT.115 (Nano Temper Technologies). Kd values were calculated as the average of the triplicate experiments analyzed with MO.Affinity Analysis 3 software.

### 2.5. Dynamic Light Scattering

hGCAP1 were concentrated to 175 µM in 20 mM Tris-HCl pH 8, 5 mM 2-mercaptoethanol, and filtered through a 0.22 µm filter (Merck Millipore Ltd. Burlington, MA, USA). Eighty microliters of protein solution were used to measure hydrodynamic radius by DLS. Measurements were performed using a DynaPro instrument (Protein Solutions, Charlottesville, VA, USA) in a thermostatic cuvette at 10 °C.

### 2.6. SEC-MALS

Size-Exclusion Chromatography (SEC) combined with Multi-Angle Light Scattering (MALS) detection was performed in an HPLC system composed by a Waters 515 HPLC Pump, a Waters 2487 Dual λ Absorbance detector (Waters, Sesto San Giovanni, Italy), a Wyatt Dawn Heleos MALS and a Wyatt Optila T-rEX differential refractive index detector (Wyatt Technology, Santa Barbara, CA, USA).

200 μL of 2 mg/mL GCAP1 samples were run on a Superdex 75 Increase 10/300 GL column (GE Healthcare, Milan, Italy), by using 20 mM Tris-HCl, pH 8, 5 mM 2-mercaptoethanol, with/without 1 mM CaCl_2_, as mobile phase, at a flow rate of 0.8 mL/min. Molar masses were calculated by means of the Astra V software vs. 5.3.4.20 (Wyatt), using a dn/dc value of 0.185.

### 2.7. Trp Fluorescence

The Trp fluorescence of the hGCAP1 was measured diluting the protein concentration at 3 µM in the followed three different buffers: calcium buffer (20 mM Tris-HCl, 150 mM KCl, pH 8, 5 mM DTT, 1 mM CaCl_2_), magnesium buffer (20 mM Tris-HCl, 150 mM KCl, pH 8, 5 mM DTT, 1 mM MgCl_2_, 1 mM EGTA), EGTA buffer (20 mM Tris-HCl, 150 mM KCl, pH 8, 5 mM DTT, 1 mM EGTA). All the measurements were performed in a Varian Cary Eclipse fluorescence spectrophotometer using a 280 nm excitation wavelength, 300–400 nm emission wavelength, setting the slit at 10 nm, and the temperature at 25 °C.

### 2.8. SAXS Data Collection

The SAXS data of hGCAP1 were collected with λ = 0.9919 Å after elution from the analytic SEC (Superdex75 Increase 10/300 GL) at ESRF beamline BM29 with Pilatus 1M detector with 1 s exposure time.

The 2700 frames of hGCAP1 in EGTA (20 mM Tris-HCl pH 7.5, 150 mM NaCl, 5 mM EGTA, 5 mM DTT) were collected at 20 °C: the selected peak comprised 47 frames (1372–1419) that were scaled in groups of three and subtracted from the buffer (frames 15–194) considering each slightly different concentration (from UV absorbance) before subtraction.

The SEC-SAXS data of hGCAP1 in Mg^2+^ (20 mM Tris-HCl pH 8.0, 5 mM MgSO_4_, 2.5 mM EGTA, 150 mM KCl, 5mM DTT) and Ca^2+^/Mg^2+^ (20 mM Tris-HCl pH 8.0, 5 mM CaCl_2_, 5 mM MgSO_4_, 150 mM KCl, 5 mM DTT) were collected at 4 °C. The selected peak of hGCAP1 in Ca^2+^/Mg^2+^ at the higher protein concentration comprised 41 frames (1330–1371, buffer frames 172–525). The data analysis was performed using programs PRIMUS 3.0 [18] and ScÅtter 3 (http://www.bioisis.net)).

### 2.9. Model of hGCAP1 Dimeric Assembly and Molecular Dynamics Simulations

Different possible dimeric conformations were generated using the equilibrated structure of Ca^2+^-loaded hGCAP1 reported in [19] as the monomeric unit. The structure was subjected to three independent rigid body docking simulations using ZDOCK 3.0.2 [20] with a sampling step of 6° (dense sampling) starting from different relative orientations, each resulting in 4000 complexes. All the 12,000 resulting dimers were then filtered according to the distance constraints identified by Lim et al. [15]. The standard deviation of such constraints was increased by 50% to include correctly the distances between Cαs and the limits of rigid-body docking, resulting in the following constraints: 42.5 Å < d (E57–E57) < 57.5 Å; 42.5 Å < d (E133–E133) < 57.5 Å, and, 23.5 Å < d (E154–E154) < 32.5 Å. The 3 final dimers, resulting from the highest-scored filtered complex from each of the three docking runs, were then fused to a randomly generated C-terminal (17 amino acids).

The three dimers (d1, d2, and d3) were then subjected to 200 ns all-atom MD simulations using GROMACS v. 2016.1 [21] with CHARMM36m [22] as a force field, after manual addition of the parameters for the myristoylated Gly at the N-term [23]. The simulated systems consisted of a dodecahedral box (~11.5 × 11.5 × 8.2 nm^3^) comprising 30673–33644 water molecules, 93–96 K^+^ ions, 13–14 Mg^2+^ ions, and 95–101 Cl^−^ ions for a total of 98566–107500 atoms, depending on the specific dimer. The protocols for energy minimization, equilibration, and production were the same as in Marino et al. [4].

For each trajectory, 2000 frames were extracted at 100 ps intervals and the average χ^2^ with respect to SAXS data over the last 100 ns of the trajectory was calculated using the program CRYSOL 2.8 [24] both considering and discarding the C-term. Indeed, the presence of the highly mobile C-term was found to greatly affect the agreement with SAXS data making it difficult for the selection of the correct dimer interface. To include in the analysis the mobility of each dimeric assembly along with the simulations we used the program EOM 2.1 [25] (with *saxns_fit2eom* script; http://xray.utmb.edu/SAXNS) that builds a mixed model selecting different structures along the simulations to enhance the agreement with SAXS data.

From each trajectory, the frame displaying the lowest absolute χ^2^ without C-ter (i.e., 162 for d1, 1878 for d2, and 1921 for d3) was selected as the representative model for each dimer, namely d1′, d2′, and d3′, respectively.

In such dimers, the two chains were separated and subjected again to rigid-body docking using ZDOCK 2.3 [26]. For each complex, 4 docking simulations were performed (1 with the original relative orientation and 3 with randomly generated orientations) each with 4.000 solutions and rotational sampling 6°. The 16,000 solutions for each complex were clustered into a group of native-like poses having a Cα RMSD < 1 Å with respect to the original complex. The average ZDOCK score (ZD-s) was then calculated for the native-like poses to estimate the binding free energy using the correlation explained in [27] (ΔG^0^ = 3.86–0.39 × ZD-s).

Finally, in silico mutagenesis of the variants known to interfere with the dimerization process, namely H19R, Y22D, F73E, and V77E [15], on both monomers of the best complexes (d1′ and d2′) were obtained using Maestro (Schroedinger) software. The rotamers for each residue were selected according to the most probable non-clashing rotamer ranked by the “mutate residue” function of Maestro. The four variants were docked using ZDOCK 2.3 [26], using the same protocol as previously described for the WT homodimers, and the relative change in binding free energy with respect to the WT (ΔΔG^0^) was calculated. The persistence of electrostatic interactions over the 200 ns trajectories was calculated using PyInteraph 1.0 software [28], which calculates the percentage of the trajectory frames where the distance constraints between oppositely charged residues are fulfilled (3.5 Å).

## 3. Results

### 3.1. hGCAP1 Expression and Purification

The recombinant hGCAP1 was expressed in E. coli BL21-CodonPlus RP cells with N–terminal myristoylation (Myr). In such conditions, we expected an efficient cut of the N-ter Met (>97%) [29] followed by an effective N–terminal myristoylation (>95%). The protein was purified from the soluble fraction of the cell lysate by ion-exchange chromatography followed by size exclusion chromatography. The purification yield was typically 10 mg (protein)/g (cells).

### 3.2. Thermal Denaturation of hGCAP1

Thermal denaturation of hGCAP1 with and without Ca^2+^ was detected using circular dichroism spectroscopy. The ellipticity at 220 nm of the Ca^2+^/EGTA protein varied from –26/–27 mdeg (at 20 °C) to –19/−22 mdeg (at 95 °C) (Appendix A). In both cases, the persistence of the ellipticity signal at high temperature demonstrates a non-complete denaturation of the protein and the conservation of most of the alpha-helical structures. Such evidence suggests that the EF-hand motifs could remain well folded and functional even at extreme conditions [30].

### 3.3. Monomer-Dimer Equilibrium of hGCAP1 in the Presence of Ca^2+^ or Mg^2+^

We used analytical size exclusion chromatography (SEC) to study the monomer-dimer equilibrium of hGCAP1 (molecular weight, Mw = 22.9 kDa). Comparing the elution volume (Ve) of the Ca^2+^ bound hGCAP1 (Ve = 10.89 mL) with that of 5 reference proteins (from Aprotinin, Mw = 6.5 kDa to Conalbumin, Mw = 75 kDa) it was possible to estimate its molecular weight at 49 kDa, compatible with a dimeric assembly. In these experimental conditions, the hGCAP1 concentration at the elution peak, calculated from UV absorbance at 280 nm (Mw/ε = 1.07 mg/mL), was 17.7 μM. Reducing the protein concentration, the SEC peak shifted toward higher Ve (i.e., lower apparent Mw), suggesting the presence of a monomer-dimer equilibrium.

To characterize such equilibrium, we performed six analytical SEC experiments in the presence of Ca^2+^ (Figure 1A). Ve depends on protein apparent Mw which, in turn, is related to the dissociation constant (Kd) of the monomer-dimer equilibrium. Therefore, it is possible to represent the variations of Ve with protein concentration with an equation depending on 3 free parameters, including Kd (Figure 1C). Using this analysis, we estimated a Kd of 8.8 ± 0.7 μM for the calcium-bound hGCAP1 (fitting curve in Figure 1, lower left panel).

The same experiment in EGTA/Mg^2+^ buffer (Figure 1B) allowed to calculate a higher value for the dissociation constant (Kd = 45 ± 15 μM; Figure 1D). These results demonstrate that the Ca^2+^ bound protein is more prone to dimerize than the Mg^2+^ bound protein.

### 3.4. Monomer-Dimer Equilibrium of hGCAP1 Analysed with Microscale Thermophoresis

We performed additional experiments to determine the hGCAP1 dimer dissociation constant using microscale thermophoresis. We assumed that the labeled protein (with NT-647-NHS fluorescent dye) at low concentration (20 nM) was monomeric. Therefore, the addition of increasing amounts of unlabeled protein-induced changes in labeled protein fluorescence depending on the formation of the dimeric assembly (Figure 2). In this way, we calculated a monomer-dimer dissociation constant for the Ca^2+^ bound hGCAP1, Kd = 9.4 ± 1.3 µM (Figure 2, blue points/curve) in agreement with the SEC results. The same experiment repeated in the presence of Mg^2+^/EGTA resulted in a Kd one order of magnitude higher (Kd = 77.2 ± 8.9 µM; Figure 2, red points/curve).

### 3.5. Monomer-Dimer Equilibrium Is Affected by Ionic Strength

Additional SEC experiments were performed varying the concentration of apo hGCAP1 (obtained as described in Materials and Methods), in a buffer with low ionic strength (20 mM Tris-HCl, 5 mM 2-mercaptoethanol, pH 8.0). These experiments showed constant values of the elution volumes, indicating the loss of the monomer-dimer equilibrium and the presence of a single quaternary assembly. Multi-angle light scattering coupled with SEC (SEC-MALS) analysis showed an Mw corresponding to the hGCAP1 monomer (Mw ~ 23 kDa; Figure 3, red curve). In low ionic strength conditions, hGCAP1 remained monomeric even at high concentration (175 μM), as shown in dynamic light scattering (DLS) experiments that estimated an Mw of 24 kDa (18% polydispersion).

Interestingly, the addition of a low amount of calcium (1 mM) to the salt-free buffer was sufficient to induce again the monomer-dimer equilibrium, with a protein molecular weight calculated from MALS of about 29 kDa (Figure 3, blue curve). The slightly higher hydrodynamic radius of the apoprotein (Figure 3, red curve) can be explained by intramolecular repulsive effects due to the protein net charge in the salt-free buffer and by the increased exposition of hydrophobic surface that requires a thicker solvation shell [30].

An analogous result was also observed for chicken GCAP1 (data not shown) demonstrating that the binding of calcium is a determinant factor to promote GCAP1 dimerization.

### 3.6. hGCAP1 Trp Fluorescence in the Different Ion Binding States

hGCAP1 hosts 2 tryptophan residues: Trp21, in the Myr cavity, and Trp94, between EF2 and EF3 hand motifs. The Trp fluorescence measured in the presence of Mg^2+^ or EGTA showed similar values of maximum emission wavelength (343.5 and 343.3 nm, respectively). Such values are correlated with significant solvent exposure of tryptophan, as discussed by Vivian and Callis [31], that showed a variation range of maximum emission wavelength between 310 and 350 nm for completely buried or completely exposed Trp, respectively. In the presence of saturating Ca^2+^ concentration (5 mM) we registered a 2 nm redshift of the emission peak (345.2 nm), indicating that Ca^2+^ binding slightly increases the solvent exposure of Trp (Appendix A). Such evidence suggests a slightly higher hydrodynamic radius of the Ca^2+^-bound form in agreement with the lower elution volumes observed in SEC experiments (Figure 1, upper panels). Similar behavior was observed for bovine GCAP1 which showed a change from 345 to 347 nm upon Ca^2+^ addition [32]. Such structural changes induced in GCAP1 by calcium-binding are likely correlated with the promotion of the dimeric assembly as observed for the Ca^2+^ bound protein with respect to the Mg^2+^ bound moiety.

### 3.7. SAXS Analysis of the hGCAP1 Monomer

The SEC-SAXS analysis of hGCAP1 at low concentration (10.9 µM) (analyzed using programs PRIMUS 3.0 [18] and ScÅtter 3 (http://www.bioisis.net/)) without Ca^2+^ (20 mM Tris-HCl pH 7.5, 150 mM NaCl, 5 mM EGTA, 5 mM DTT) showed a Mw, estimated from scattering intensity relative to water at zero angle (I_0_ = 18.29 ± 0.06), of 21.1 kDa [33], compatible with a monomeric assembly (Rg = 21.7 ± 0.1 Å; Figure 4A). After regularization and real space transformation (program GNOM; ATSAS 3.0.1 [34]; Dmax = 60 Å; Figure 4B) the superposition of a typical low resolution model (program DAMMIN 5.3 [35]; Figure 4, upper panel)) on cGCAP1 crystal structure (pdb-id: 2r2i; Dmax ~ 50 Å; Figure 4C) shows that hGCAP1 occupies a slightly larger volume. Anyway, the modest agreement of cGCAP1 with the SAXS data (program CRYSOL [24]; χ^2^ = 2.62) was greatly improved by the addition of the missing 17 C-terminal residues (with program CORAL 1.1 [36]; χ^2^ = 1.43; Figure 4D,E).

### 3.8. SEC-SAXS Evidence of the Monomer-Dimer Equilibrium

We collected SEC-SAXS data of hGCAP1 in different buffers and at different protein concentrations. The SEC absorbance peaks were used to estimate protein concentrations to properly scale the scattering data before averaging and buffer subtraction. By comparing the scattering data extrapolated at zero angles of protein and water as a reference, it is possible to estimate the protein Mw (Table 1). As discussed above, apo hGCAP1 (in EGTA) is monomeric at low concentration (10.9 μM). On the contrary, when bound to Ca^2+^ or Mg^2+^, hGCAP1 evolves toward different monomer-dimer equilibria whose Kd depends on the binding state of the EF-hands motifs, as already described.

In particular, Ca^2+^ bound protein displays an Mw varying from 35.2 kDa (at 25.3 μM) to 43.7 (at 51.1 μM).

The SEC-SAXS data of calcium bound GCAP1 collected at the higher protein concentration (~76 uM; Table 1), corresponding to a prevalent dimeric assembly (Rg = 24.5 ± 0.1 Å), were regularized and transformed in real space with the program GNOM (ATSAS 3.0.1) [34]; real space Rg = 24.8 ± 0.1 Å, Dmax = 80 Å; Figure 5A), and then used to produce several low resolutions models using DAMMIF (ATSAS 3.0.1) with P2 symmetry ([37]; ensemble resolution = 29 ± 2 Å; Figure 5C). These data were also used to select between different possible dimeric assemblies as described in the next paragraph.

### 3.9. Modelling and Selection of Possible Dimeric Assemblies

The three-dimensional structural model for hGCAP1 (201 amino acids, 22.9 kDa) was obtained by homology modeling using the X-ray structure of cGCAP1 (pdb-id: 2r2i, sequence identity of 84%; [38]) as a template. Analysis of the possible protein-protein interfaces hosted in the cGCAP1 crystallographic packing of the chicken protein with the PDBePISA server [39] did not reveal any stable dimeric assembly. However, previous work [15] based on DEER provided structural hints as to the dimeric nature of bovine GCAP1 and defined specific distance-constraints among residues belonging to each monomer, resulting in two alternative dimeric assemblies a symmetric and an asymmetric one.

In view of these lines of evidence, the homology modeled and equilibrated structure of hGCAP1 described by Marino and Dell’Orco [19] was used to produce 12,000 random dimers with rigid body docking using the program ZDOCK 3.0.2 [20]. The dimers were then filtered according to the distance constraints between the 4 residues identified by Lim et al., [15] i.e., His19, Tyr22, Phe73, and Val77 (see Methods) resulting in 36 to 43 filtered poses for each independent docking run obtained by randomizing the initial position of the probe. The three assemblies presenting the highest ZDOCK 3.0.2 score in each independent and randomized docking run (p1, p2, and p3) were significantly different from each other (Cα-RMSD values were: 4.48 Å for p1–p2, 14.49 Å for p2–p3 and 12.73 Å for p1–p3). The three best-scored dimers (from now on named dimers d1, d2, and d3) were then fused with a randomly generated C-terminus and subjected to 200 ns all-atom MD simulations sampling a total of 2000 frames (one every 100 ps).

Firstly, we analyzed the agreement of the identified dimers along each respective MD trajectory omitting the highly mobile C-terminal tail (i.e., the last 17 amino acids) of every monomer. The average χ^2^ calculated on the final 50 ns of the MD simulations (Appendix A) shows that dimers d2 and d3 display a comparable agreement with scattering data (<χ^2^> = 2.79 and 3.71, respectively), whereas d1 exhibits a poorer agreement (<χ^2^> = 13.45). The ranking changes when considering in the models also the conformations of C-ter, with d1 and d2 being in comparable agreement with SAXS (<χ^2^> = 3.02 and 3.40, respectively) and d3 in disagreement (<χ^2^> = 13.79) (Appendix A). To take into account also the dynamics of the proteins along with the 3 simulations (including the C-ter) we used the program EOM 2.0 (Ensemble Optimisation Method; [25]) that builds a mixed/dynamical model for every simulation with the weighed contribution of different structural states to fit the SAXS data. Again, the dynamical analysis confirmed d1 and d2 as the best assemblies (χ^2^ = 1.16 and 1.17, respectively) and d3 as the worst (χ^2^ = 2.37) (Figure 5B).

To summarize, the agreement with SAXS data of d1 and d2 improves by adding the C-ter amino acids and it gets even better when protein dynamics are explicitly considered. On the contrary d3—the best dimer in the absence of C-ter—worsens considerably when taking into account the contribution of the C-ter end.

For each MD simulation, the structural conformation showing the lowest χ^2^ with respect to the SAXS data (without C-ter; Table 2) (named d1′,d2′,d3′) was subjected again to rigid-body docking using another version of the ZDOCK algorithm (ZDOCK 2.3) to obtain an estimate of the free energy of binding, using an empirical correlation with the ZDOCK score as explained in [27]. This approach has demonstrated to be useful to predict in silico the effects of any point mutation affecting the dimer interface on the relative free energy of binding (ΔΔG^0^), under the assumption that mutations do not significantly perturb the structure of each interacting molecule. The scores of SAXS and ZDOCK analysis are reported in Table 2.

### 3.10. Computational Analysis of Mutations in hGCAP1 Dimeric Assembly

All the dimers predicted by the rigid-body docking algorithm accounting for shape and electrostatic complementarity and desolvation of residues buried at the dimer interface (ZDOCK 2.3) resulted in high-score poses, and 16–24 of the 12,000 poses reconstituted within 1 Å (Cα-RMSD; “native-like” poses; Table 2) with respect to dimers d1′,d2′,d3′ (Table 2). As it has been proven that, in the absence of major conformational changes in either of the interacting protein, a correlation exists between the average docking score of native-like poses and the standard free energy of binding, we performed in silico mutations to generate the variants shown to disrupt the dimer and shift the equilibrium toward the monomer, namely F73E, H19R, V77E, and Y22D [15]. Interestingly, for all of the three dimers, the mutations resulted in a lower number of native-like poses still fulfilling the DEER distance constraints (Table 3). The computational analysis of the 12 mutagenized homodimers (ZDOCK 2.3; Table 3), shows for all the structures a shift toward the monomeric form, in line with the experimental results [15], though to a different extent (Table 3). Indeed, for every assembly, the H19R variant has a greater effect on the stability of the dimer. The predicted effects of the four point mutations on the ΔΔG^0^ values are high for all the dimers but are especially apparent for d1′, for which a 4.5 to 5.8 kcal/mol effect was predicted, which would strongly shift the equilibrium toward the monomeric form.

Considering together the agreement with SAXS data and the overall Z-DOCK results we can select d1 and d2 as the best dimeric assemblies with the lower “dynamical” χ^2^ vs. SAXS data, the highest docking score, and the higher perturbing effect of the mutations on the dimeric assembly. The good agreement of the d1 dynamical model with SAXS data is further shown in the superposition with one of the low-resolution DAMMIF models (NSD 1.67; Figure 5C).

### 3.11. Insights into the Molecular Interface of the Most Likely Dimers d1 and d2

The different orientation between the monomers in the alternative dimeric assemblies d1′ and d2′ is shown in Figure 5D, evidencing a rotation of about 60°. Molecular dynamics simulations showed that the interface between dimers d1 and d2 involved persistent electrostatic interactions between the negatively charged residues of the EF2 loop involved in Ca^2+^-coordination, namely D68, D72, and E75, and the positively charged residues of the entering helix of EF3 (Appendix A), namely R93 and K97, although with peculiarities for each dimer. Specifically, dimer d1 exhibited persistent interactions between R93 and D68 (70.6% of the simulation time), R93 and D72 (18.3%), and R93 and E75 (18.7%). Such switching electrostatic interactions hinged on R93 were completely absent in dimer d2, which on the other hand showed electrostatic interactions hinged on K97 (with D68 (7.1%), D72 (72.9%), and E75 (18%)), also negligible in d1.

When an accurate analysis of the static interface of dimers d1′ and d2′ was performed by Bioluminate (version 3.9.072, Schroedinger), a slightly increased buried solvent accessible surface area for d2′ (245.1 Å^2^) with respect to d1′ (219.4 Å^2^) with a similar shape complementarity score (2487 vs. 2541) were measured. Interestingly, F73 was found to be involved in a π-stacking interaction in both interfaces, specifically with Y22 in dimer d1′ and with F65 in dimer d2′. Finally, the static analysis confirmed the electrostatic interactions involving D72 specifically with R93 in d1′ (salt bridge) and with K97 in d2′ (charge-reinforced H-bond), which could contribute to the highest docking score (ZD-s) of the first dimeric assembly (Table 2). In conclusion, both static and dynamic analyses of the dimeric interfaces show a balance of stabilizing interactions, that does not allow to define which one among d1′ and d2′ is the most representative hGCAP1 dimer.

## 4. Discussion

SAXS data on hGCAP1 at a low concentration (10.9 µM) without Ca^2+^ or Mg^2+^ are compatible with a monomeric protein whose structure is in good agreement with the crystal structure of chicken GCAP1 as long as the C-ter end is included in a loose conformation (causing the increase of the radius of gyration from 19.8 to 21.0 Å).

Purified hGCAP1 analyzed with different experimental techniques shows the equilibrium between monomer and dimer. Since the protein has a theoretical isoelectric point of 4.4 and is negatively charged at physiological pH, we expect a prevailing effect of electrostatic repulsions at low salt concentration shifting the equilibrium towards the monomeric state, as observed in SEC-MALS experiments. The electrostatic repulsion is partially shielded by binding one/two Mg^2+^ ions or, to a greater extent, by three Ca^2+^ ions. Indeed, we observed that at low salt concentration the addition of a small amount (1 mM) of calcium is sufficient to push the monomeric state toward the monomer-dimer equilibrium. In conditions closer to the physiological state, i.e., at medium salt concentration and in the Ca^2+^-bound state, hGCAP1 displays a monomer-dimer equilibrium with Kd~9 μM. When the 3 calcium ions are exchanged with one/two Mg^2+^ the higher electrostatic repulsion screen decreases the dimerization Kd (~60 μM).

Interestingly, also the chicken protein, despite being monomeric in the crystal structure, displays a similar dynamic monomer-dimer equilibrium (not shown), showing that this property is common to different GCAP1 orthologs.

The shift toward dimeric assembly favored in the presence of Ca^2+^ can also explain the protein resistance to trypsin with respect to the unbound state observed in previous works [41,42].

The differences in monomer-dimer equilibrium suggest conformational changes induced by GCAP1 ligation state as observed with Trp fluorescence experiments, that are likely related to GCAP1 regulatory activity on retGC1, as suggested by thorough MD simulations [19].

Using a combination of computational techniques, we produced 3 models of reliable dimeric interfaces compatible with the restrains reported by Lim et al., [15]. In every model, the dimeric interface is based on the N-terminal domain of GCAP1, already known to be involved in both protein dimerization and the interaction with retGC1 [6]. Our docking simulations confirmed the crucial role of the residues identified in bovine GCAP1 by Lim et al. [15] in stabilizing the dimeric assembly, as their in silico substitutions resulted in a dramatic increase of ΔΔG^0^ values, compatible with a shift of hGCAP1 to its monomeric form. It is interesting to highlight that the higher destabilization is associated with the substitution involving H19 for both d1′ and d2′ (Table 3). The H19Y mutation was recently identified in patients diagnosed with retinitis pigmentosa [13]. Indeed, the mutant showed a profound shift in Ca^2+^-sensitivity of RetGC regulation and, in line with our present observations, its oligomeric equilibrium did not shift to the dimer in the presence of Ca^2+^, thus suggesting a correlation between GCAP1 physiological role and its dimerization.

## 5. Conclusions

Monomer–dimer equilibrium in proteins is determined by a subtle balance between stabilizing and destabilizing forces [43]. Based on our models, only a small accessible surface area is buried on dimer formation (~9% of the total protein surface). Such surface is nonetheless sufficient to create strong electrostatic and shape complementarity between the monomers and suggests a high contribution of the hydrophobic effect to the process of desolvation of the protein–protein interface, which is reflected in the high docking scores (Table 2). The hydrophobic contacts at the GCAP1 dimer interface have been proven to be essential for both its dimerization and for the activation of RetGC, which is itself a dimer, thus suggesting that the GCAP1 dimer may bind to the cyclase target to form a 2:2 complex, whose allosteric regulation of catalytic activity may involve quaternary structural changes in a protein-protein complex [16]. An alternative explanation for the role of bGCAP1 dimerization has been proposed by Lim et al. [15]. In the absence of bRetGC, the dimeric nature of bGCAP1 in the photoreceptor inner segment may prevent its diffusion into the outer segment. The partial overlap of the bGCAP1-bRetGC and bGCAP1-bGCAP1 interface suggests that, as soon as retGC becomes available, a tight intermolecular interaction would occur, which allows the mature RetGC/GCAP1 complex to incorporate in transport vesicles that are then transported to the outer segments, where the complex could then translocate into disk membranes [15]. However, the actual physiological role of GCAP1 dimerization in human photoreceptors is not completely clarified and is only partially compatible with such a hypothesis. Assuming that the intracellular concentration of hGCAP1 is similar to that of bGCAP1 in bovine rod outer segments [44], namely 3.3 μM, our experimentally determined Kd values would lead to ~33% of hGCAP1 being dimeric at high Ca^2+^, and only ~9% at low Ca^2+^, in the Mg^2+^-bound state. The apparent affinity of hGCAP1 for hRetGC (3.2 μM, see [30] is 3-fold higher than the affinity of dimerization in the presence of Ca^2+^ (9 μM) and 19-fold higher than that in Mg^2+^ (~60 μΜ), therefore hGCAP1 would rather bind to its target regulator than to itself under both conditions. Dimerization of hGCAP1 could then rather serve as a regulatory mechanism to prevent an excessive diffusion to the outer segment in the absence of RetGC, and not necessarily a condition to achieve a 2:2 complex. An updated estimate of the intracellular concentration of both GCAP1 and retGC in human photoreceptors is therefore necessary to clear up the actual molecular scenario.

GCAP1 structural changes related to the presence of Ca^2+^ or Mg^2+^ are reflected in the observed changes of the dimer dissociation constant. Such conformational changes are likely related to the mechanisms of GCAP1 to control the activity of the retGC1 partner, and therefore their investigation here described represents a step forward to dissect the structural bases of the regulatory mechanism of GCAP1.

## Figures and Tables

**Figure 1 biomolecules-10-01408-f001:**
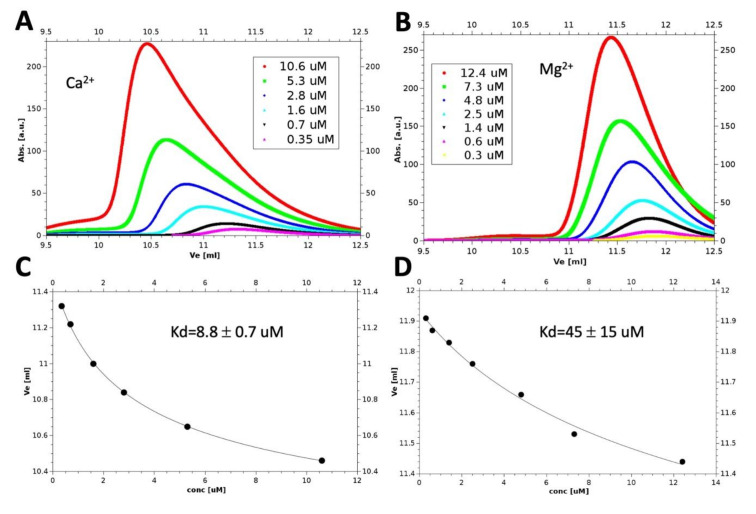
Dissociation constants calculated from size exclusion chromatography (SEC) experiments. SEC experiments in presence of Ca^2+^ (**A**) or Mg^2+^ (**B**) at different protein concentrations. In the lower panels (**C**,**D**) are reported the corresponding variations of elution volumes (Ve) with protein concentration (black dots) together with the 3-parameters theoretical curves fitting the experimental data (black lines).

**Figure 2 biomolecules-10-01408-f002:**
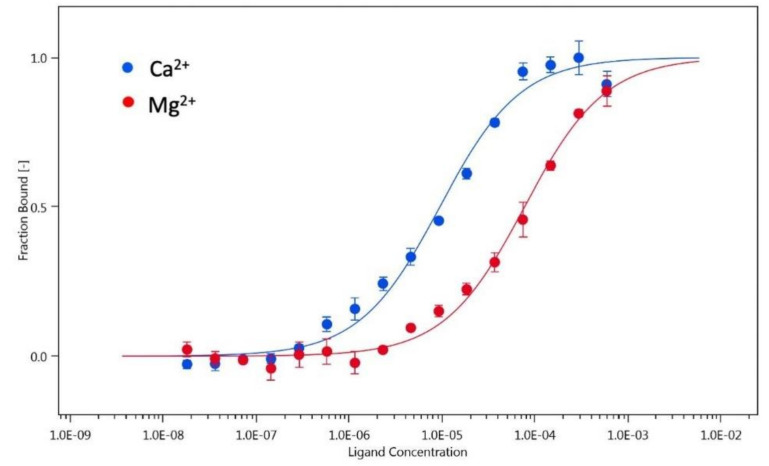
Microscale thermophoresis of the hGCAP1 monomer-dimer transition in Ca^2+^ and Mg^2+^. Changes in labeled protein fluorescence as a function of unlabeled hGCAP1 concentration in the presence of Ca^2+^ (blue points) or Mg^2+^ (red points) together with the curves fitting the two experiments (law of mass action) in blue and red, respectively.

**Figure 3 biomolecules-10-01408-f003:**
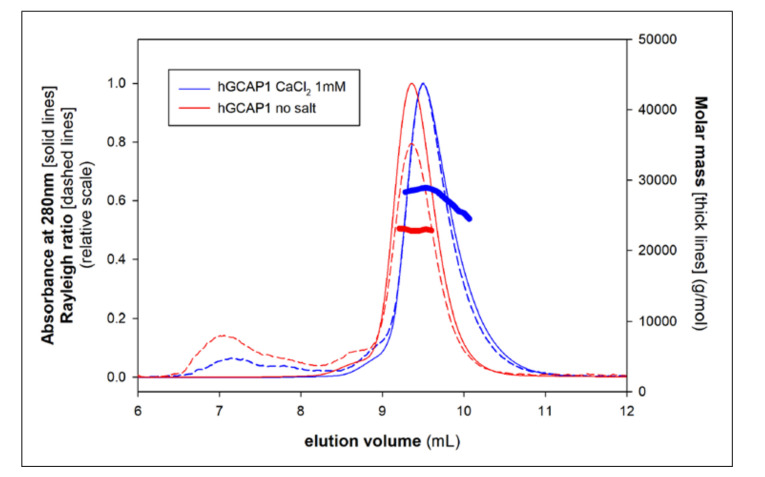
Multi-angle light scattering coupled with SEC (SEC-MALS) of hGCAP1 in the salt-free buffer. Without salt hGCAP1 is monomeric (Mw 23 kDa; red curves) and it moves toward the monomer-dimer equilibrium upon the addition of 1 mM Ca^+2^ (Mw ~29 kDa, blue curves). The thick lines represent the molar mass measured by MALS (legend on the right) in correspondence to each SEC peak (solid lines; legend on the left); the dashed lines represent the scattering signal (legend on the left).

**Figure 4 biomolecules-10-01408-f004:**
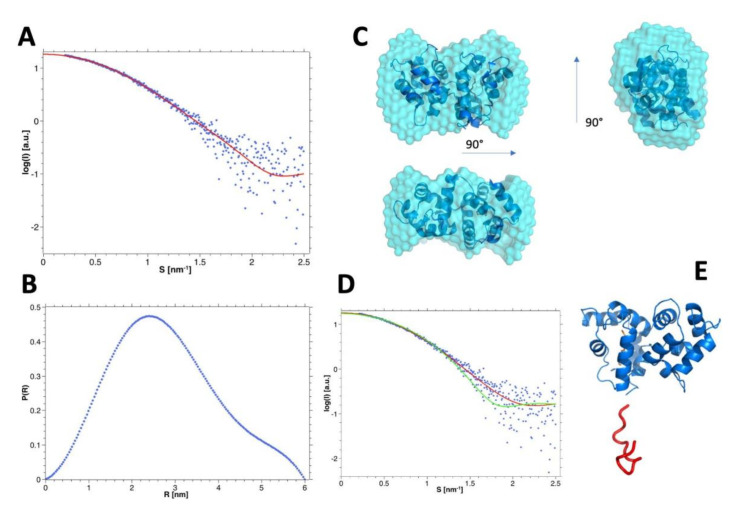
SEC-SAXS data of hGCAP1 monomer. (**A**) experimental SEC-SAXS data of hGCAP1 without Ca^2+^ or Mg^2+^ at 10.9 μM (Rg = 21.5 Å; blue points) with GNOM regularization (red line) and (**B**) corresponding P(R) (Dmax = 60 Å). (**C**) DAMMIN most typical low-resolution model (cyan surface; ensemble resolution = 25 ± 2 Å; superposed to the crystal structure of cGCAP1 (NSD 2.09; pdb-id 2r2i; Rg = 19.8 Å; blue cartoons). (**D**) experimental data fitted with cGCAP1 without C-ter (CRYSOL χ^2^ = 2.62; green curve) or with C-ter (added with CORAL; red curve; χ^2^ = 1.43). (**E**) hGCAP1 model obtained adding to the chicken protein (blue cartoons) the C-ter with coral (red cartoon). The pictures of protein structures were prepared with the PyMOL 2.2.0 program (The PyMOL Molecular Graphics System, Version 2.0 Schrödinger, LLC).

**Figure 5 biomolecules-10-01408-f005:**
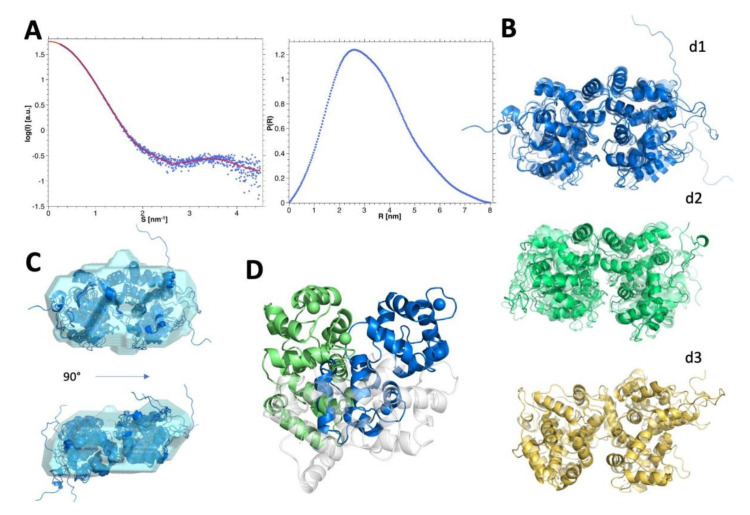
SEC-SAXS data of hGCAP1 dimer. (**A**) GNOM fitting (red line) of the SEC-SAXS data (blue points) at the higher protein concentration and corresponding P(R). (**B**) Dynamical EOM models of the 3 dimeric assemblies—d1-d3—in cartoons with transparency depending on the occupation of each structure on the assembly. (**C**) Superposition of the dynamical model d1 (blue cartoon) on one of the most representative low-resolution models (DAMMIF, P2 symmetry, ensemble resolution = 29 ± 2 Å, NSD 1.67) depicted as the cyan surface in 2 different orientations. (**D**) Different orientation of the monomers d1′—blue cartoons and d2′—green cartoons in the static dimeric assemblies (the superposed monomer is shown as white transparent cartoons).

**Table 1 biomolecules-10-01408-t001:** SEC-SAXS parameters for hGCAP1 in different conditions.

Buffer	Conc. [μM] ^a^	Rg [Å]	I_0_	Mw (I_0_) ^c^ [kDa]	% mon.	Dmax [Å]	Mw (Vc) [kDa]	Mw (q_max_) [kDa]
EGTA	10.9	21.5	18.2	21.1	100	60	29.4	28.2
Mg^2+^	59.4	22.4	32.3	35.9	43	75	29.0	28.8
Ca^2+^/Mg^2+^	25.3	24.1	31.6	35.2	46	76	27.4	27.1
Ca^2+^/Mg^2+^	51.1	24.5	39.2	43.7	9	81	35.2	38.0
Ca^2+^/Mg^2+^	~76 ^b^	24.5	-	-	-	81	40.8	45.6

^a^ Concentrations based on peak absorbance. ^b^ This concentration was not measured directly due to experimental problems; therefore, the reported value is estimated, and it does not allow the analysis based on I_0_ values. ^c^ Mw calculated using the program *PRIMUS* and relative scale respect to water I_0_ (0.0156 for EGTA sample and 0.0162 for the other samples).

**Table 2 biomolecules-10-01408-t002:** Agreement with SAXS data and *ZDOCK* analysis and for the 3 selected models d1′,d2′,d3′.

Assembly	‘Static’ χ^2^ with SAXS ^a^	‘Dynamic’ χ^2^ with SAXS ^b^	ZD-s *^c^*	Native-Like Poses ^c^	Best Ranked Poses ^d^
**d1′**	2.42	1.16	54.4 ± 0.8	22	1
**d2′**	1.95	1.17	44.5 ± 0.9	24	16
**d3′**	2.08	2.37	44.9 ± 0.7	16	1

^a^ χ^2^ values of selected individual structures along with the MD simulation without the C-ter. ^b^ χ^2^ values of selected mixed structures along with the MD simulation with the C-ter. ^c^ Average score of native-like poses (i.e., within 1 Å Cα-RMSD with respect to the original complex) from multiple docking runs. ^d^ Rank of the native-like pose with the highest score out of 12,000 poses.

**Table 3 biomolecules-10-01408-t003:** Effect of the mutations on Z-dock scores and prediction of the relative free energy of binding.

Assembly	Mutant	Native-Like Poses	ΔZD-s	ΔΔG^0^ * [kcal/mol]
**D1′**	F73E	15	−13.14	5.12
	H19R	12	−14.75	5.75
	V77E	21	−11.62	4.53
	Y22D	17	−12.20	4.76
**D2′**	F73E	12	−2.65	1.04
	H19R	8	−4.85	1.89
	V77E	9	−4.15	1.62
	Y22D	9	−4.70	1.84

* ΔΔG^0^ was calculated by using an empirical correlation between ZD-s and ΔG^0^ as explained in [27] and in [40].

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
