# Peer review of "Modulation of Guanylate Cyclase Activating Protein 1 (GCAP1) Dimeric Assembly by Ca2+ or Mg2+: Hints to Understand Protein Activity"

_biomolecules, 2020, doi:10.3390/biom10101408_

Round 1

Reviewer 1 Report

The manuscript presents SAX, MALS and circular dichroism data that investigate the dimerization of GCAP1 protein.  The experiments demonstrate that Ca2+ binding to GCAP1 stabilizes the dimeric form and present molecular structures of Ca2+ bound GCAP1 dimer.  Overall, the experiments and analysis seem satisfactory and the conclusions are supported by the data.  I have only one question that should be addressed:  Why does Ca2+-bound GCAP1 have a longer elution volume than metal-free GCAP1 in Fig. 3?  The Ca2+-bound GCAP1 has larger molar mass than metal-free GCAP1 and yet the elution volume of Ca2+-bound GCAP1 is longer than that of metal-free GCAP1.  There needs to be some molecular explanation of the discrepancy here.

Author Response

Question 1: Why does Ca2+-bound GCAP1 have a longer elution volume than metal-free GCAP1 in Fig. 3?  The Ca2+-bound GCAP1 has larger molar mass than metal-free GCAP1 and yet the elution volume of Ca2+-bound GCAP1 is longer than that of metal-free GCAP1.  There needs to be some molecular explanation of the discrepancy here.

Answer 1: We agree that the Figure 3 can be misleading and requires additional comments. We explain the counterintuitive result with the fact that the apo protein (lower Mw) has a slightly higher hydrodynamic radius than the calcium-bound protein (higher Mw, corresponding to the monomer-dimer equilibrium) in the salt-free buffer. Such result can be explained by the protein net charge that, in salt-free buffer, produces an intramolecular repulsive effect inducing the monomeric protein to adopt an enlarged structure i.e. higher hydrodynamic radius. An additional effect producing changes in elution volume could be due to buffer-related (i.e. with/without calcium) changes in protein-resin interaction. Moreover, some of us previously showed by ANS-fluorescence that cation-free GCAP1 has a more extended hydrophobic surface, (ref. 30), which would require a thicker solvation shell that increases the hydrodynamic radius.

Action: We added at pag. 8 rows 338-341 the sentence: "The slightly higher hydrodynamic radius of the apo protein (Figure 3, red curve) can be explained by intramolecular repulsive effects due to the protein net charge in the salt-free buffer and by the increased exposition of hydrophobic surface that requires a thicker solvation shell [30]".

Reviewer 2 Report

Boni, et. al., present a combined experimental and theoretical (modeling) study of the monomer/dimer equilibrium for guanylate cyclase activating protein 1 that provides novel insight regarding the metal binding-dependence of this equilibrium. The manuscript is well written, providing clear and concise descriptions of the work that was done, the data, and the findings that result from the data. The authors, refreshingly, do an excellent job of clearly and carefully delineating the limitations of their findings. Overall, I find this study to be of high quality and strong potential impact in understanding both the functionality of GCAMP1 and the role of oligomerization processes in metal-binding proteins. The majority of my comments are related to minor typos or grammatical corrections, but there are a few technical issues that I would like to see addressed in a minor revision prior to publication. These issues are:

  1. The authors use NT-647-NHS as the fluorescent label for the microscale thermophoresis measurements, but they do not address where this label attaches to the protein and how the labeling site relates to the dimerization interface. Given the fact that the thermophoresis data results in a considerably different Kd for the Mg-bound state than was determined from SEC measurements, I wonder if the dye has an impact on the measurement due to its localization. It would be helpful if the authors addressed this issue briefly to clarify such a concern.
  2. The SEC data presented in Figure 1 shows a clear shift for both conditions examined, but the positions of the peaks at high concentration (presumably the oligomerized form) is quite different for the Ca-bound (10.5 mL) versus Mg-bound (11.5 mL) states of the protein. The data for the MG-bound form seems to include a small signal around 10.5 mL that would overlap with the position of the dimer peak under Ca-bound conditions. Perhaps this small signal is too minor to warrant addressing or interpreting, but it struck me as interesting that it is at the same position as the Ca-bound oligomer (potentially representing a minor species or the presence of a small amount of residual Ca?) The authors do not address the relevance of the difference in position of the peaks, which I think is a more significant omission. A brief statement addressing this difference and explaining its relevance (or irrelevance) would be helpful. I wonder, for example, if the difference in position might be indicative of a chance in the compactness (or dynamics) of the protein upon Ca-binding. 
  3. I do not understand why the authors performed and discussed experiments examining the intrinsic Trp fluorescence but do not include the data in the main manuscript or in the supplemental information. The final statement of paragraph 3.6 indicates that the findings from these experiments have a likely relation to the Ca-binding promotion of dimerization, so I would like to see these data included in the manuscript or supplemental so the reader can evaluate them directly. Though a 2 nm shift is small (and perhaps near the limit of resolution), changes in the emission peak width or shape may also be informative. The authors are encouraged to include these data and expand the discussion of the data somewhat, even if both data and discussion are included in the supplemental information. 

Minor corrections:

line 44: photoreceptors should say phototreceptor's or photoreceptors'

line 126: species is misspelled

line 158: 'added or no' is unclear and should be revised to improve clarity. 

line 208: build should say builds

Figure 3: There are two lines included in the figure, one red and one blue, that appear to be drawn in? The presence of these lines is not addressed. If they are significant, they should be explained. If they are not significant, they should be removed. 

line 511: 'induced by EF-hands ligation state' is confusing and should be revised. It is not clear if the ligation state of a single EF-hand is being referenced or that of more than one. The Trp data is referenced here as being insightful, so additional clarity would be gained by showing and discussing those data more thoroughly. 

line 518: 'resulted in in dramatic' should be corrected.

Line 544: should read 'compatible with such a hypothesis'

Author Response

Reviewer 2:

Question 1: The authors use NT-647-NHS as the fluorescent label for the microscale thermophoresis measurements, but they do not address where this label attaches to the protein and how the labeling site relates to the dimerization interface. Given the fact that the thermophoresis data results in a considerably different Kd for the Mg-bound state than was determined from SEC measurements, I wonder if the dye has an impact on the measurement due to its localization. It would be helpful if the authors addressed this issue briefly to clarify such a concern.

Answer 1: We thank the reviewer for noticing this missing information. The NT-647-NHS carries a reactive NHS-ester group that reacts with primary amines (lysine residues which are usually solvent accessible) to form a covalent bond, achieving a 1:1 ratio of labeled protein to dye. Since hGCAP1 contains 11 Lys and from our dimeric models just Lys97 could be involved in the dimerization interface we conclude that the possible perturbation of the dimeric interface statistically will occur 1/11 of the times, producing an overall effect that should be negligible in the estimation of the differences in Kd for calcium or magnesium bound protein. Definitely the Kd values measured with microscale thermophoresis experiments substantially agree with those calculated from gel filtration experiments considering the experimental errors and the possible small perturbation due to Lys labelling. Such agreement is already summarized in the discussion reporting an average value (~60 μM) for Kd of the Mg2+ bound protein: “the higher electrostatic repulsion screen decreases the dimerization Kd (~60 μM)”.

Action: in M&M we added the sentence: “The NT-647-NHS carries a reactive NHS-ester group that reacts with primary amines (lysine residues) to form a covalent bond, achieving about 1:1 ratio of labeled protein to dye.”

Question 2: The SEC data presented in Figure 1 shows a clear shift for both conditions examined, but the positions of the peaks at high concentration (presumably the oligomerized form) is quite different for the Ca-bound (10.5 mL) versus Mg-bound (11.5 mL) states of the protein. The data for the MG-bound form seems to include a small signal around 10.5 mL that would overlap with the position of the dimer peak under Ca-bound conditions. Perhaps this small signal is too minor to warrant addressing or interpreting, but it struck me as interesting that it is at the same position as the Ca-bound oligomer (potentially representing a minor species or the presence of a small amount of residual Ca?).

Answer 2: We thank the reviewer for the interesting notation: in our opinion the small signal around 10.5 mL of the Mg-bound form corresponds to aggregation states analogously to the small bump around 9.8 mL in the Ca-bound form. Such negligible aggregation states are often present in other experiments. Action: none

Question 3:The authors do not address the relevance of the difference in position of the peaks, which I think is a more significant omission. A brief statement addressing this difference and explaining its relevance (or irrelevance) would be helpful. I wonder, for example, if the difference in position might be indicative of a chance in the compactness (or dynamics) of the protein upon Ca-binding.

Answer 3: We agree with the reviewer remark: the difference in peak position must be (also) related to differences in overall hydrodynamic radius due to the different binding state of the EF-hands of the protein. The presence of the high dynamical monomer-dimer equilibrium makes the two observables, i.e. monomer/dimer equilibrium and monomer dimension, difficult to be separated. Anyway, some hints can be derived by the Trp fluorescence experiments that show a slightly increased solvent exposure of Ca2+-bound form suggestive of a slightly higher hydrodynamic radius in agreement with SEC peak position.

Action: we added to the Trp fluorescence paragraph the sentence: "Such evidence suggests a slightly higher hydrodynamic radius of the Ca2+-bound form in agreement with the lower elution volumes observed in SEC experiments (Figure 1, upper panels).

Question 4: I do not understand why the authors performed and discussed experiments examining the intrinsic Trp fluorescence but do not include the data in the main manuscript or in the supplemental information. The final statement of paragraph 3.6 indicates that the findings from these experiments have a likely relation to the Ca-binding promotion of dimerization, so I would like to see these data included in the manuscript or supplemental so the reader can evaluate them directly. Though a 2 nm shift is small (and perhaps near the limit of resolution), changes in the emission peak width or shape may also be informative. The authors are encouraged to include these data and expand the discussion of the data somewhat, even if both data and discussion are included in the supplemental information. 

Answer 4: We thank you the reviewer for this comment: we did not include any figure because of the slight variation 2 nm.

Action: we added a new figure in ‘supplemental materials’

Question 5: Minor corrections:

line 44: photoreceptors should say phototreceptor's or photoreceptors'

line 126: species is misspelled

line 158: 'added or no' is unclear and should be revised to improve clarity. 

line 208: build should say builds

Answer 5: we corrected the text as suggested.

Question 6: Figure 3: There are two lines included in the figure, one red and one blue, that appear to be drawn in? The presence of these lines is not addressed. If they are significant, they should be explained. If they are not significant, they should be removed. 

Answer 6: The thick lines represent the molar mass measured by MALS (legend on the right) in correspondence of each SEC peak (solid lines, legend on the left); the dashed lines represent the scattering signal (legend on the left).

Action: we added this information to the figure legend. Figure 3. SEC-MALS of hGCAP1 in salt-free buffer. Without salt hGCAP1 is monomeric (Mw 23 kDa; red curves) and it moves toward the monomer-dimer equilibrium upon addition of 1 mM Ca+2 (Mw ~29 kDa, blue curves). The thick lines represent the molar mass measured by MALS (legend on the right) in correspondence of each SEC peak (solid lines; legend on the left); the dashed lines represent the scattering signal (legend on the left).

Question 7: line 511: 'induced by EF-hands ligation state' is confusing and should be revised. It is not clear if the ligation state of a single EF-hand is being referenced or that of more than one. The Trp data is referenced here as being insightful, so additional clarity would be gained by showing and discussing those data more thoroughly.

Answer 7: in the sentence we are referring to the whole protein: to clarify the point we changed the sentence: "The differences in monomer-dimer equilibrium suggest conformational changes induced by GCAP1 ligation state as observed with Trp fluorescence experiments, that are likely related to GCAP1 regulatory activity on retGC1, as suggested by thorough MD simulations [19]".

Questions 8:

line 518: 'resulted in in dramatic' should be corrected.

Line 544: should read 'compatible with such a hypothesis'

Answer 8: we corrected the text as suggested.